# The Protective Effects of Lactoferrin on Aflatoxin M1-Induced Compromised Intestinal Integrity

**DOI:** 10.3390/ijms23010289

**Published:** 2021-12-28

**Authors:** Ya-Nan Gao, Song-Li Li, Xue Yang, Jia-Qi Wang, Nan Zheng

**Affiliations:** 1Key Laboratory of Quality & Safety Control for Milk and Dairy Products of Ministry of Agriculture and Rural Affairs, Institute of Animal Sciences, Chinese Academy of Agricultural Sciences, Beijing 100193, China; gyn758521@126.com (Y.-N.G.); lisongli@caas.cn (S.-L.L.); yangxue234723@126.com (X.Y.); jiaqiwang@vip.163.com (J.-Q.W.); 2Laboratory of Quality and Safety Risk Assessment for Dairy Products of Ministry of Agriculture and Rural Affairs, Institute of Animal Sciences, Chinese Academy of Agricultural Sciences, Beijing 100193, China; 3Milk and Milk Products Inspection Center of Ministry of Agriculture and Rural Affairs, Institute of Animal Sciences, Chinese Academy of Agricultural Sciences, Beijing 100193, China; 4State Key Laboratory of Animal Nutrition, Institute of Animal Sciences, Chinese Academy of Agricultural Sciences, Beijing 100193, China

**Keywords:** aflatoxin M1, lactoferrin, intestinal barrier, tight junction, transcriptome, proteome

## Abstract

Aflatoxin M1 (AFM1), the only toxin with maximum residue levels in milk, has adverse effects on the intestinal barrier, resulting in intestinal inflammatory disease. Lactoferrin (LF), one of the important bioactive proteins in milk, performs multiple biological functions, but knowledge of the protective effects of LF on the compromised intestinal barrier induced by AFM1 has not been investigated. In the present study, results using Balb/C mice and differentiated Caco-2 cells showed that LF intervention decreased AFM1-induced increased intestinal permeability, improved the protein expression of claudin-3, occludin and ZO-1, and repaired the injured intestinal barrier. The transcriptome and proteome were used to clarify the underlying mechanisms. It was found that LF reduced the intestinal barrier dysfunction caused by AFM1 and was associated with intestinal cell survival related pathways, such as cell cycle, apoptosis and MAPK signaling pathway and intestinal integrity related pathways including endocytosis, tight junction, adherens junction and gap junction. The cross-omics analysis suggested that insulin receptor (INSR), cytoplasmic FMR1 interacting protein 2 (CYFIP2), dedicator of cytokinesis 1 (DOCK1) and ribonucleotide reductase regulatory subunit M2 (RRM2) were the potential key regulators as LF repaired the compromised intestinal barrier. These findings indicated that LF may be an alternative treatment for the compromised intestinal barrier induced by AFM1.

## 1. Introduction

As an excellent source of bioactive protein, milk and dairy products are indispensable in the daily diet for people of all ages, providing essential nutrients and supporting human health [1]. Human exposure to aflatoxin M1 (AFM1) is mainly through the consumption of contaminated milk that comes from dairy cows that have ingested aflatoxin B1 (AFB1)-contaminated feed. Due to its heat stability, AFM1 is regarded as a potential risk for human health worldwide, especially for infants and young children due to their high consumption of dairy products [2,3]. To decrease the health risk caused by AFM1, many countries and organizations have established a maximum residue level (MRL) for AFM1 in milk, varying from 0 to 1000 ng/L, with 50 ng/L set by EU and 500 ng/L set by China and the United States [2]. The challenges of global presence of AFM1 in milk were well summarized in a recent study, indicating that products with high AFM1 concentrations which markedly exceeded 500 ng/L were found in Africa, Asia, Europe and South America [4]. It has been classified into Group 1 human carcinogen [5]. Previous studies demonstrated that AFM1 caused a compromised intestinal barrier, due to decreased tight junction (TJ) proteins and mucin expression as well as induced inflammation [6,7,8,9,10]. Thus, considerable effort has been made to investigate detoxification approaches for AFM1 in milk around the world. However, the data regarding the protective effects of nutritional compounds on AFM1-induced intestinal barrier are limited.

Lactoferrin (LF) is an important bioactive protein in mammalian milk, especially in the colostrum, and its characteristics have been reviewed recently [11]. In varied species, LF is highly conserved with amino acid homology between human LF (hLF) and bovine LF (bLF) exceeding 70%, and also exhibiting some identical biological functions in various experimental model [11]. The US Food and Drug Administration (FDA) has approved bLF for practical use as a supplement for infant formula, as it is generally recognized as safe (GRAS) [11].

Considering that bLF is known for its antibacterial, anti-inflammatory and immunoregulatory properties [12], the reported studies regarding the protective effects of bLF on gut health focus on the prevention of inflammation and modulation of intestinal microbiota [13,14]. Most research has concentrated on the direct effects of bLF in the model of normal cells or healthy animals, where one study showed that bLF could increase the expression TJ proteins and enhance the intestinal epithelial barrier in human intestinal epithelial crypt cells (HIECs) and Caco-2 cells [15]. There are also several studies demonstrating the anti-inflammatory activity of LF, even in patients suffering from Crohn’s disease [16,17,18,19]. Although our previous reported study demonstrated that bLF could inhibit AFM1-induced DNA damage in human intestinal Caco-2 cells [20], the underlying mechanisms of the positive effects of LF on a compromised intestinal barrier, especially on intestinal integrity still needs to be explored.

TJ proteins are area multi-protein complexes that are essential components in regulating the intestinal integrity, pivotal to maintaining body health [21]. In the present study, it was hypothesized that LF intervention could reduce AFM1-induced injury on intestinal integrity. To verify this hypothesis, individual LF and AFM1 as well as their combination were exposed to Balb/C mice and differentiated Caco-2 cells to evaluate the intestinal barrier function. In addition, cross-omics analysis of transcriptome and proteome was used to explore the underlying key regulators of the LF protective effects. These results will provide a new insight into the preventive effects of LF on AFM1-induced intestinal barrier dysfunction.

## 2. Results

### 2.1. LF Inhibited AFM1-Induced Compromised Intestinal Barrier In Vivo

Compared with the control and DMSO group, mice body weight did not change significantly (*p* > 0.05) in the AFM1 and LF groups during the experiment, recorded on days 4, 10, 16, 22 and 28 (Figure 1A). The results of serum indicators related to intestinal barrier integrity indicated that AFM1 damaged the intestinal barrier, with significantly (*p* < 0.05) depressed concentration of citrulline (Cit) and increased levels of intestinal fatty acid binding protein (I-FABP) and D-lactate. The addition of LF significantly restored the Cit level and partly decreased the I-FABP and D-lactate contents in the AFM1 treatment group (Figure 1B–D). The histological morphology exposed to AFM1 and LF treatment was shown in Figure 2A–E. Compared with the control group, shorter and thicker villi were observed in the AFM1 treatment group. The injured ileum was markedly improved by LF with increased villus length (V), reduced crypt depth (C) and increased V/C ratio (*p* < 0.05), as seen in Figure 2F,G. In the control group, the structure of TJ proteins, such as occludin, claudin-3 and ZO-1 were intact in a clear grid-like pattern, and the immunostaining intensity was strong (Figure 3). The TJ protein structure in AFM1 exposure was collapsed, which was reversed by LF treatment (Figure 3).

### 2.2. Effect of LF on Injured Differentiated Caco-2 Cells Induced by AFM1

Compared with the control group, AFM1 significantly decreased trans-epithelial electrical resistance (TEER) values (*p* < 0.05) in differentiated Caco-2 cells, while it was partly reversed by the LF treatment, although there was no marked difference in the comparison with LF + AFM1/AFM1 (Figure 4A). From the paracellular flux results of lucifer yellow (LY), it can be seen that consistent with TEER values, AFM1 induced an increasingly differentiated Caco-2 cell permeability. Compared with AFM1 individually, the combination of LF and AFM1 significantly suppressed the paracellular passage of LY, improving the intestinal barrier function (*p* < 0.05), as seen in Figure 4B.

### 2.3. Effect of LF on the Gene Expression in AFM1 Treatment In Vitro

The transcriptome analysis was conducted in differentiated Caco-2 cells exposed to AFM1 and LF treatment individually and in combination. Compared to the control group, there were 4993 differentially expressed genes (DEGs) upon AFM1 treatment, of which 1898 were up- and 3095 were downregulated; 280 in the LF treatment group, of which 88 were up- and 192 were downregulated; and 4193 in the LF + AFM1 treatment group, of which 1854 were up- and 2339 were downregulated, as seen in Figure 5A. The results of principle component analysis (PCA) showed that no differences occurred between the control and LF treatments, or between AFM1 and LF + AFM1 treatment (Figure 5B). This indicated that LF appeared to have a slight effect on differentiated Caco-2 cells and AFM1 played a decisive role in the combination of LF and AFM1. The Venn results showed that compared with individual AFM1 and LF treatments, there were 562 unique DEGs in the combination treatment with LF and AFM1 (Figure 5C).

The intestinal barrier-related Kyoto Encyclopedia of Genes and Genomes (KEGG) pathways were conducted in Figure 5D, with individual and combined AFM1 and LF treatment affecting the intestinal cell-related signaling pathways, such as apoptosis, cell cycle, MAPK and Wnt signaling pathway, and the intestinal integrity-related pathways, including regulation of actin cytoskeleton, focal adhesion, adherens junction, gap junction, and tight junction to varying degrees. However, the number of DEGs in these related pathways in the AFM1 group was more than in the combination group. The results of quantitative reverse transcription polymerase chain reaction (qRT-PCR) analysis were consistent with the transcriptome analysis, confirming the transcriptome reliability (Figure 5E). There were no significant changes (*p* > 0.05) between control and individual LF treatment in these selected genes’ expression level, which was consistent with the transcriptome results that these selected genes were not DEGs between control and individual LF treatment. While as the transcriptome results, the mRNA expression levels between control and AFM1 or LF + AFM1 treatment were significantly altered (*p* < 0.05; Figure 5E). The top 20 KEGG pathways, enriched by the 562 unique DEGs in the combined LF and AFM1 treatment, included intestinal barrier related pathways, such as mucin type O-glycan biosynthesis, notch signaling pathway and ECM-receptor interaction (Appendix A).

### 2.4. Effect of LF on the Protein Expression in AFM1 Treatment In Vitro

The proteome analysis was conducted in differentiated Caco-2 cells exposed to AFM1 and LF treatment individually and in combination. The number of differentially expressed proteins (DEPs) in the AFM1 treatment group were 471, of which 166 were up- and 305 were downregulated; LF had 76, of which 16 up- and 60 downregulated; and LF + AFM1 had 406, of which 136 were up- and 270 were downregulated (Figure 6A). The PCA analysis of the proteome was consistent with that of the transcriptome, indicating that AFM1 had a more obvious effect than LF (Appendix A). The results of proteomic Venn results showed that 228 unique DEPs were exerted in the combination of LF and AFM1 treatment, compared with individual AFM1 and LF treatment (Figure 6B). The AFM1, LF and LF + AFM1 treatments affected the intestinal barrier-related pathways, including apoptosis, p53 and Wnt signaling pathway in different degrees (Figure 6C). The intestinal integrity-related pathways, such as regulation of actin cytoskeleton, adherens junction, gap junction, tight junction and focal adhesion, participated in the AFM1 and LF + AFM1 treatments (Figure 6C). For those affected pathways, LF individual treatment and the combined treatment were least affected, while AFM1 treatment had the greatest influence on them. The inflammation-related pathways, including the IL-17 signaling pathway and Inflammatory bowel disease (IBD), were only enriched in the AFM1 treatment (Figure 6C). The protein expression of CYP1A1 and PCK1 were measured by Western blotting, which was consistent with the results of the proteome (Figure 6D). Consistent with the proteomic results, compared with control group, the protein expression level significantly decreased (*p* < 0.05) in AFM1 and LF + AFM1 treatment at (Figure 6D). The intestinal integrity-related pathways were also enriched by the unique DEPs in the LF + AFM1 treatment, including the PI3K-Akt signaling pathway, endocytosis, ECM-receptor interaction, regulation of actin cytoskeleton, Rap1 signaling pathway, cell adhesion molecules, and adherens junction (Appendix A), and tight junction as well as Toll-like receptor signaling pathway (data not shown).

### 2.5. Omics-Analysis of the Transcriptome and Proteome

The scatter-plot analysis of log2-transformed ratios of mRNA: protein for LF, AFM1 and their combination is shown in Figure 7. This study concentrated on the analysis of the quadrants c and g, the significant consistent changes in both mRNA and protein levels. In the individual LF group, there were two proteins labeled as red points (FC > 2 and *p* < 0.05) in quadrants c and g, while they were no related with intestinal integrity. If the terms of the protein selection labeled gray points (FC > 2 and *p* > 0.05) are eases, there were 23 proteins as the sum of red points and gray points in quadrants c and g, but none have a relationship with intestinal integrity (Figure 7A). For AFM1 individually, quadrants c and g included 207 proteins labeled as red points and 38 proteins among them were related with intestinal integrity, including endonuclease G (ENDOG), myosin heavy chain 14 (MYH14), glycogen synthase kinase 3 beta (GSK3B), family member 3 (SMAD3), and myosin light chain kinase (MYLK) belonging to apoptosis, regulation of actin cytoskeleton, focal adhesion, adherens junction and tight junction, respectively. In addition, 52 of 333 gray points in AFM1 treatment in quadrants c and g were relevant to intestinal integrity (Figure 7B). For the combination of LF and AFM1, there were 179 proteins as red points in quadrants c and g, and among these proteins, 48 proteins were associated with the regulation of actin cytoskeleton, p53 signaling pathway, complement and coagulation cascades and tight junction, such as ribonucleotide reductase regulatory subunit M2 (RRM2), integrin subunit alpha E (ITGAE), clusterin (CLU) and claudin 3 (CLDN3). Forty-two of the 298 gray points in the LF + AFM1 treatment in quadrants c and g were also relevant to intestinal integrity (Figure 7C). To further analyze the proteins in quadrants c and g, there were 16 shared proteins related to intestinal integrity between AFM1 and LF + AFM1 treatment (Table 1). Compared with AFM1, the expression of insulin receptor (INSR), cytoplasmic FMR1 interacting protein 2 (CYFIP2) and dedicator of cytokinesis 1 (DOCK1), belonging to regulation of actin cytoskeleton, focal adhesion and adherens junction were higher in the combination of LF and AFM1 treatment group, while the expression of RRM2 in the p53 signaling pathway was lower.

## 3. Discussion

As a first barrier to defense against exogenous threats, the intestinal barrier plays a significant role in human health, especially for newborns, who have an immature gastrointestinal system [22]. The compromised intestinal integrity and especially increased intestinal permeability, has been proven to be associated with various intestinal diseases, including food allergies, Crohn’s disease, and inflammatory bowel diseases [23,24,25]. Numerous studies have theorized that food components could assist the compromised intestinal barrier caused by mycotoxins, such as zearalenone (ZEN), T-2 toxin and deoxynivalenol (DON) [26,27,28,29]. It is reasonable to speculate that LF, the vital bioactive protein in mammalian milk, could protect the disrupted intestinal barrier induced by AFM1 in vivo and in vitro.

In the present study, the addition of LF did not significantly alter the mice body weight treated by AFM1 in vivo compared with AFM1 individually (*p* > 0.05, Figure 1A). However, LF significantly increased the concentration of Cit, decreased the concentration of D-lactate and I-FABP (*p* < 0.05, Figure 1B–D) in the serum of mice, indicating that LF improved the injured intestinal barrier induced by AFM1. These clinical serum parameters could reflect the state of the intestinal barrier [30,31,32]. The level of D-lactate was significantly enhanced in broiler chicks, which were fed with 1.5 mgAFB1/kg diet [33]. However, they have not been widely used in assessing the dietary effects induced by mycotoxins and LF; thus, there are only a few studies reported [15,20].

The results of examinations of mice intestine structures also demonstrated the protective effects of LF on AFM1 in Figure 2 and Figure 3. These results were consistent with a study that founded that lactoferricin B, a component derived from lactoferrin, could attenuate the intestinal barrier disruption induced by Escherichia coli O157:H7 in mice [34]. It has been observed that LF intervention increased the villi height of the jejunum and the protein expression of occludin of jejunum and ileum in suckling piglets [35]. LF also increased the ratio of villus length/crypt depth in the mice exposed to radiation through downregulating NF-κB expression, reducing the intestinal injury [36].

The results in differentiated Caco-2 cells also showed that LF reversed AFM1-induced increased intestinal permeability (Figure 4), consistent with the in vivo findings in the present study. It has been reported that LF ameliorated the increased permeability induced by tumor necrosis factor α (TNF-α) in HT-29/B6 monolayers [37]. In addition, LF reversed the rat and human intestinal IEC-18 and Caco-2 cell damage caused by hydrogen peroxide by increasing cell viability and decreasing inflammation [38]. The use of LF could enhance human intestinal epithelial cell proliferation by inhibiting the MAPK and NF-κB signaling pathways, or by activating the PI3K/Akt signaling pathway [39,40].

The underlying mechanisms of the protective effects of LF were clarified by the cross-omics data of the transcriptome and proteome. For the cross-omics analysis, the decreased fold change for the key genes PKC, TJP1, ACTB-G1 and MYLK and proteins MYH and SMAD3 associated with intestinal integrity in AFM1 was stronger than for the LF + AFM1 treatment. The expression of these genes and proteins was significantly increased in the AFM1 and ochratoxin A (OTA) treatment [6]; thus, it can be speculated that AFM1-induced intestinal barrier dysfunction was partly reversed by LF.

This study found that the protective effects of LF in intestinal barrier dysfunction were associated with two types of signaling pathways, including one related with the survival rate of epithelial cells and the other is the intestinal integrity. The pathways associated with cell viability included cell cycle, apoptosis, mTOR signaling pathway, MAPK signaling pathway, p53 signaling pathway, and PI3K-Akt signaling pathway. All of the above pathways have been reported to be involved in the process by which LF stabilizes the intestinal barrier function [36,37,39]. In addition, endocytosis, tight junction, adherens junction, gap junction, focal adhesion and ECM-receptor interaction participated in intestinal integrity. It has been reported that the intestinal integrity dysfunction induced by AFB1 and AFM1 was related to clathrin-mediated endocytosis [41]. An impaired intestinal barrier may result in the intestinal inflammatory diseases [42,43], and this study found that the inflammation related pathways, such as the IL-17 signaling pathway were only enriched by the DEGs and DEPs upon AFM1 treatment.

Combining the analysis of transcriptome and proteome, this study found that INSR, CYFIP2, DOCK1 and RRM2 played a critical role in the protective effects of LF on AFM1-induced intestinal barrier dysfunction (Table 1). It has been demonstrated that insulin resistance played an important role in the protective effects of decaffeinated coffee on intestinal function [44]. It was also reported that DOCK4 was associated with the repair of the intestinal barrier dysfunction caused by chemical material, and a positive correlation was exerted between DOCK4 and mucin 2, the necessary substances for maintaining the intestinal barrier [45].

In conclusion, the present study for the first time demonstrated that in vivo and in vitro LF intervention could alleviate AFM1-induced intestinal barrier dysfunction. The protective effects of LF on AFM1 were related to the intestinal cells and intestinal integrity related pathways. More specifically, INSR, CYFIP2, DOCK1 and RRM2 participated in the protective effects of LF. These findings provide a new insight into the potential beneficial properties of LF, which can be used as a functional food to restore intestinal epithelial injury.

## 4. Materials and Methods

### 4.1. Chemicals

The AFM1 standard was obtained from Pribolab (Qingdao, China). BLF was purchased from Sigma-Aldrich (St Louis, MO, USA). The content of Cit, I-FABP, and D-lactate in mouse serum were detected by enzyme linked immunosorbent assay (ELISA) kits, which were purchased from SinoGene Scientific (Beijing, China). Primary antibodies of claudin-3, occludin, ZO-1, β-actin and the secondary antibody rabbit IgG conjugated to horseradish peroxidase were obtained from Bioss (Beijing, China). The dilution buffer of the primary antibody and secondary antibody as well as blocking buffer were purchased from Beyotime Biotechnology (Shanghai, China).

### 4.2. Animals

Fifty-four-week-old male Balb/C mice (20 ± 2 g) were obtained from Beijing Vital River Laboratory Animal Technology Co., Ltd. (Beijing, China). Mice were kept at 21 to 23 °C and 50 ± 5% humidity with a 12 h day and night cycle. Five groups with 10 mice per group were created for: gavage with physiological saline (control group), gavage with 1% DMSO/99% corn oil solution (DMSO group), gavage with individual 60.0 mg/kg b.w. LF (LF group), gavage with individual 3.0 mg/kg b.w. AFM1 (AFM1 group), and gavage with combined treatment (LF + AFM1 group). The LF and AFM1 were dissolved in physiological saline and DMSO in corn oil (1% solution), respectively.

The study followed the Chinese guidelines for animal care. The mice were gavaged with 0.2 mL/day for four weeks and individually weighted every three days. At the day 29, CO_2_ was used to euthanize these mice. The mice ileum was collected and placed in 10% formalin buffer. The retro-orbital plexus method was used to gather the mice blood samples. With the permission code of “IAS2019-3 (Date: 18/3/2019)”, the Ethics Committee of the Chinese Academy of Agricultural Sciences (Beijing, China) approved these animal experiments.

### 4.3. Serum Biochemical Indicators Determination

The serum indicators of Cit, I-FABP and D-lactate used in the present study represented the state of the intestinal barrier function. To obtain the serum, the collected mice blood samples without anticoagulant were coagulated at 37 °C for 1–2 h, and then centrifuged at 4000 rpm for 10 min at 4 °C. The serum parameters were measured by ELISA assay following the manufacturers’ introductions.

### 4.4. Histological Assessment of the Ileum in Mice

The mice ileum samples stored in 10% buffered formalin were washed with water, followed by dehydration with alcohol, then embedded in paraffin and finally cut into 5 μm sections. Hematoxylin and eosin (HE) staining for these sections was used to perform histopathological evaluation. Villus height and crypt depth of ileum were measured from 5 villi in each slide.

### 4.5. Immunofluorescence Analysis of the Ileum in Mice

The sections of mice ileum were dewaxed, underwent heat-induced antigen retrieval and then incubated with the primary antibody of occludin, claudin-3 and ZO-1 overnight at 4 °C and the corresponding secondary antibody. The nuclei were counterstained by 4′,6-diamidin-2-phenylindol (DAPI). An LSM780 immunofluorescent microscope (Carl Zeiss, Inc., Thornwood, NY, USA) was used to obtain the fluorescent images.

### 4.6. Cells Culture, TEER Values and Permeability Measurement

Caco-2 cell line (passages between 15 and 35) were purchased from the American Type Culture Collection (ATCC) (Manassas, VA, USA). As our previous study, the model of differentiated Caco-2 cells was established [7]. The 100 μg/mL LF dissolved in physiological saline and 8 μg/mL AFM1 dissolved in methanol individually and in combination added to the apical compartments of transwell chambers. After 48 h incubation, TEER values and LY permeability were assessed as the reported study [7].

### 4.7. Transcriptomics Studies

The transcriptome was performed as previous study [6]. In brief, to reduce the variability of samples, triplicate wells of differentiated Caco-2 cells were combined into one sample. For each treatment, a total of three samples were prepared. In brief, the RNA of the differentiated Caco-2 cells without LF and AFM1 treatment (control) and the cells treated with individual 100 μg/mL LF and 8 μg/mL AFM1, and their combination for 48 h were extracted. After extracting and purifying the total RNA, the library of cDNA was constructed and sequenced on an Illumina HiSeq™ 2500. With the criterion of false discovery rate (FDR) < 0.05 and fold change > 1.5, the significantly DEGs were identified. Gene Ontology (GO) and KEGG enrichment analyses were then conducted and qRT-PCR was used to validate the DEGs. The qRT-PCR was performed as the manufacturer’s instructions. Briefly, Fast Quantity RT Kit (TIANGEN, Beijing, China) was used to reverse-transcribe extracted RNA into cDNA. Primers for the evaluated genes are shown in Appendix A. The relative changes of evaluated genes were calculated as the 2^−∆∆CT^ method.

### 4.8. Data-Independent Acquisition-Based (DIA) Proteomics Studies

To reduce the sample variability, three replicates of harvested differentiated Caco-2 cells were pooled into one sample, and a total of three samples were prepared from each treatment. Differentiated Caco-2 cells without treatment were chosen as the control group. Cells treated with 100 μg/mL LF and 8 μg/mL AFM1 individually as well as their combination for 48 h were considered the treated group. After digesting the extracted protein samples of differentiated Caco-2 cells, peptides could be obtained and from the six fractions were extracted and analyzed by low pH nano LC-MS/MS. The peptides were resuspended with 30 μL solvent A (A: 0.1% formic acid), and then analyzed by LC-MS/MS equipped with an online nanoelectrospray ion source. The Orbitrap Fusion Lumos mass spectrometer connected with an EASY-nLC 1200 system (Thermo Fisher Scientific, MA, USA) was used in the present study. A total of 3 μL samples were loaded into the75 μm × 25 cm analytical column (cclaim PepMap C18) with a gradient of 5–35% solvent B (B: 0.1% formic acid) in 120 min. The column flow rate and electrospray pressure were controlled at 200 nL/min and 2 kV, respectively. Subsequently, DIA method was performed to analyze proteomics data processing. As the criteria of *p* < 0.05 and fold changes > 1.5, significantly DEPs were defined. The annotation of DEPs was conducted by GO and KEGG enrichment analysis.

Western blotting was performed to validate the DEPs. In brief, after lysing the differentiated Caco-2 cells, 100 μg protein was separated by SDS-PAGE, followed by blocking, incubating with TJ proteins’ primary antibodies and appropriate secondary antibodies. The obtained band densities were analyzed by the software of Image J 2× (Version 2.1.0, National Institutes of Health, Bethesda, MD, USA, 2006).

### 4.9. Statistical Analysis

The experimental data were analyzed by GraphPad Prism 8.0, and a one-way analysis of variance (ANOVA) and t test were used to conduct the statistical analysis. Data were represented as the mean ± standard error of the mean (SEM). A value of *p* < 0.05 was significant.

## Figures and Tables

**Figure 1 ijms-23-00289-f001:**
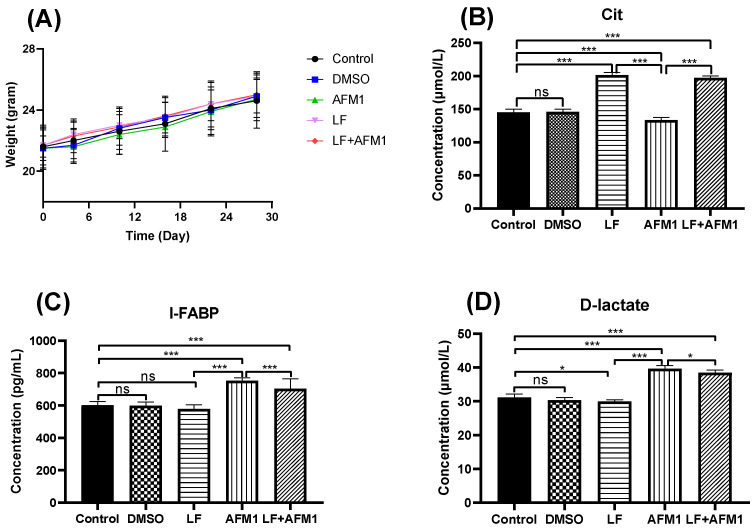
The effects of LF and AFM1 on mice body weight (*n* = 10 animals) and serum indicators. (**A**) Body weight, (**B**) Cit, (**C**) I-FABP and (**D**) D-lactate. Control represents the mice fed with physiological saline. DMSO represents the mice fed with the solution of 1% DMSO/99% corn oil. AFM1 represents the mice exposed to individual AFM1. LF represents the mice exposed to individual LF. LF + AFM1 represents the mice exposed to combined LF and AFM1. Values represent the mean ± SEM. * represents *p* < 0.05, *** represents *p* < 0.001.

**Figure 2 ijms-23-00289-f002:**
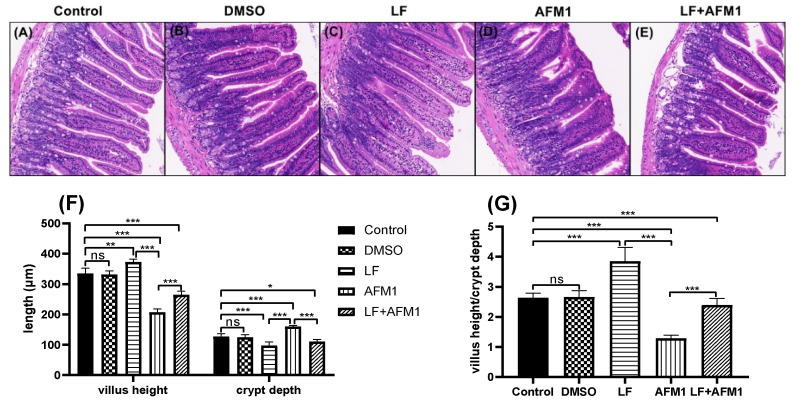
The effects of LF and AFM1 on mice ileum histology. Histology of the ileum was assessed by hematoxylin-eosin staining (HE, 200×). (**A**) Control mice (physiological saline), (**B**) DMSO-treated mice (1%DMSO/99% corn oil), (**C**) LF-treated mice, (**D**) AFM1 treated mice, (**E**) LF + AFM1 treated mice, (**F**) Villus height and crypt depth in mice ileum, (**G**) the ratio of villus height/crypt depth. Values represent the means ± SEM (*n* = three animals). * represents *p* < 0.05, ** represents *p* < 0.01, *** represents *p* < 0.001.

**Figure 3 ijms-23-00289-f003:**
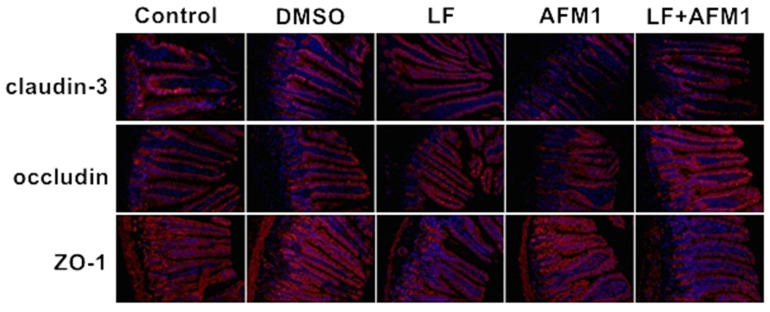
The effects of LF and AFM1 on claudin-3, occludin and ZO-1 expression and distribution in ileum measured by immunofluorescence staining (200×). Control represents the mice fed with physiological saline. DMSO represents the mice fed with the solution of 1% DMSO/99% corn oil. AFM1 represents the mice exposed to individual AFM1. LF represents the mice exposed to individual LF. LF + AFM1 represents the mice exposed to combined LF and AFM1. *n* = 3 animals.

**Figure 4 ijms-23-00289-f004:**
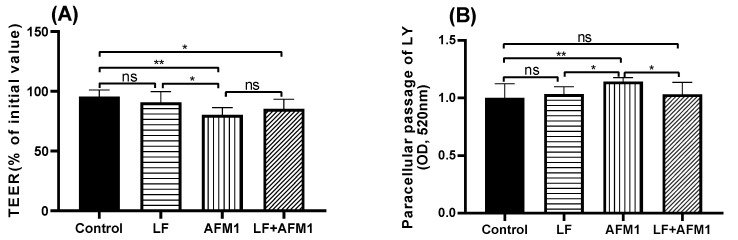
The effects of LF and AFM1 on intestinal permeability in differentiated Caco-2 cells. (**A**) TEER assay, (**B**) LY paracellular permeability. Data represent the mean ± SEM of three independent experiments (*n* = 4). Control represents the differentiated Caco-2 cells exposed to culture medium. LF represents the differentiated Caco-2 cells exposed to individual LF. AFM1 represents the differentiated Caco-2 cells exposed to individual AFM1. LF + AFM1 represents the differentiated Caco-2 cells exposed to combined LF and AFM1. * represents *p* < 0.05, ** represents *p* < 0.01.

**Figure 5 ijms-23-00289-f005:**
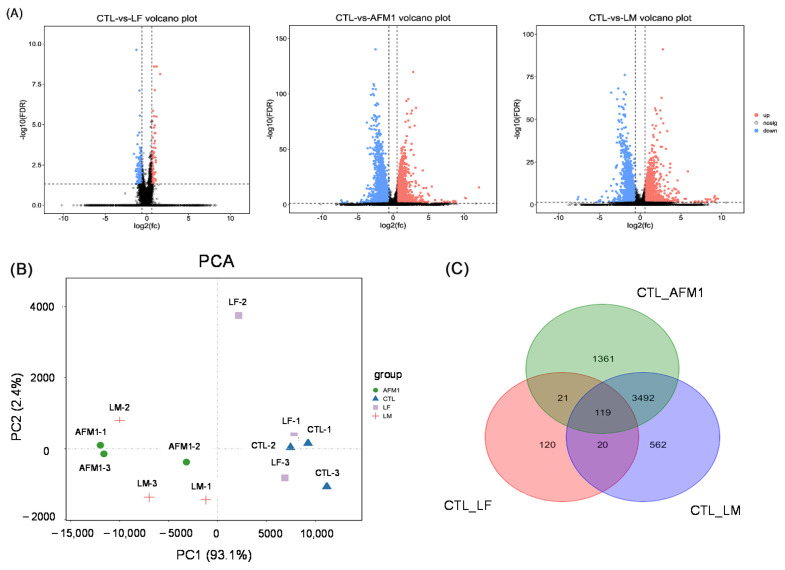
Transcriptome analysis of differentiated Caco-2 cells exposed to LF and AFM1. (**A**) The number of differentially expressed genes (DEGs). (**B**) Unsupervised PCA analysis. (**C**) Venn diagram depicting the DEGs regulated by LF and AFM1 treatment. (**D**) The intestinal integrity related KEGG pathways enriched by DEGs. (**E**) The gene expression level measured by qRT-PCR and RNA-seq. These selected genes were not DEGs between Control and individual LF treatment. LF represents individual 100 μg/mL LF, AFM1 represents individual 8 μg/mL AFM1, LM represents the combination of 100 μg/mL LF and 8 μg/mL AFM1. * represents *p* < 0.05, ** represents *p* < 0.01, *** represents *p* < 0.001.

**Figure 6 ijms-23-00289-f006:**
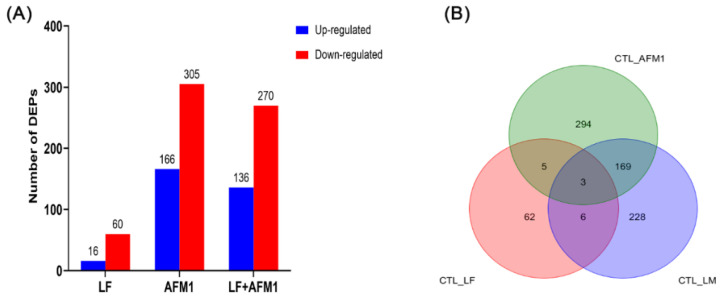
Proteome analysis of differentiated Caco-2 cells exposed to LF and AFM1. (**A**) The number of differentially expressed proteins (DEPs). (**B**) Venn diagram depicting the DEGs regulated by LF and AFM1 treatment. (**C**) The intestinal integrity-related KEGG pathways enriched by DEPs. (**D**) The protein expression level measured by Western blotting and expressed as the mean ± SEM of three independent experiments. LF represents individual treatment with 100 μg/mL LF, AFM1 represents individual treatment with 8 μg/mL AFM1, and LM represents combined treatment with of 100 μg/mL LF and 8 μg/mL AFM1. ** represents *p* < 0.01, *** represents *p* < 0.001.

**Figure 7 ijms-23-00289-f007:**
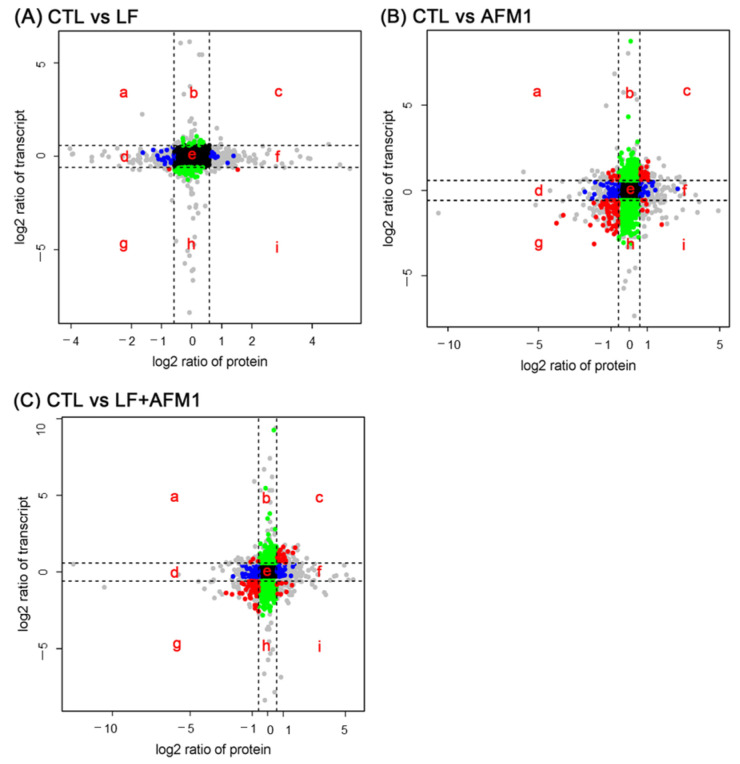
Correlation between identified transcripts and proteins in differentiated Caco-2 cells. (**A**) Cells exposed to LF, (**B**) cells exposed to AFM1 and (**C**) cells exposed to the combination of LF and AFM1. Black points (area e) represent no significant changes at both mRNA and protein levels. Blue points (area d, f) represent the significant changes at the protein level. Green points (area b, h) represent significant changes only at the transcript level. Red points (area a, c, g and i) represent significant changes in mRNA and protein levels. Gray points (area a, b, c, d, f, g, h and i) represent the transcripts or proteins meeting the criterion of *p* > 0.05 and fold change > 1.5.

**Table 1 ijms-23-00289-t001:** Correlation between DEGs and DEPs in the differentiated Caco-2 cells exposed to AFM1 and LF + AFM1 treatment.

Description	Symbol	Log2FC (AFM1/Control)		Log2FC (LM/Control)		*p* Value (AFM1/Control)		*p* Value (LM/Control)		KEGG Pathway
		Protein	mRNA	Protein	mRNA	Protein	mRNA	Protein	mRNA	
clusterin	CLU	0.67	1.32	0.77	1.14	0.0001	8.68 × 10^−53^	8.13 × 10^−5^	3.47 × 10^−33^	Complement and coagulation cascades
G protein subunit alpha i1	GNAI1	1.19	0.85	1.65	0.99	0.97	0.0001	0.65	0.0003	Rap1 signaling pathway, Chemokine signaling pathway
agrin	AGRN	−1.15	−0.88	−0.96	−0.82	1.03 × 10^−12^	9.12 × 10^−14^	9.06 × 10^−8^	7.15 × 10^−13^	ECM-receptor interaction
ribonucleotide reductase regulatory subunit M2	RRM2	−1.53	−0.99	−1.81	−1.39	8.86 × 10^−10^	0.01	5.59 × 10^−7^	0.0002	p53 signaling pathway
hepatocyte nuclear factor 4 alpha	HNF4A	−1.09	−1.57	−1.11	−1.44	1.27 × 10^−9^	7.36 × 10^−37^	2.98 × 10^−8^	1.45 × 10^−19^	AMPK signaling pathway
PVR cell adhesion molecule	PVR	−0.88	−0.91	−0.71	−0.96	3.14 × 10^−5^	9.28 × 10^−12^	0.001	3.77 × 10^−14^	Cell adhesion molecules (CAMs)
JunD proto-oncogene	JUND	−1.74	−1.71	−0.77	−1.12	0.18	1.34 × 10^−43^	0.017	5.00 × 10^−21^	MAPK signaling pathway, IL-17 signaling pathway
phosphoenolpyruvate carboxykinase 1	PCK1	−1.94	−3.14	−0.84	−0.68	0.003	1.05 × 10^−16^	0.02	2.67 × 10^−37^	FoxO signaling pathway, PI3K-Akt signaling pathway
erb-b2 receptor tyrosine kinase 3	ERBB3	−1.59	−0.90	−0.83	−0.68	0.007	1.74 × 10^−14^	0.03	1.50 × 10^−9^	ErbB signaling pathway, Calcium signaling pathway
insulin receptor	INSR	−1.29	−1.69	−1.23	−1.15	0.10	1.76 × 10^−42^	0.07	1.78 × 10^−18^	Adherens junction, HIF-1 signaling pathway
Amphiregulin	AREG	−1.08	−1.26	−1.89	−1.02	0.08	9.20 × 10^−18^	0.13	6.49 × 10^−12^	PI3K-Akt signaling pathway, MAPK signaling pathway
exocyst complex component 2	EXOC2	−0.85	−1.38	−1.19	−1.11	0.37	2.72 × 10^−23^	0.15	5.13 × 10^−14^	Ras signaling pathway
E1A binding protein p300	EP300	−1.11	−0.97	−0.93	−0.76	0.25	7.10 × 10^−11^	0.33	1.55 × 10^−6^	HIF-1 signaling pathway, Wnt signaling pathway
cytoplasmic FMR1 interacting protein 2	CYFIP2	−1.41	−1.64	−1.16	−1.31	0.02	4.49 × 10^−22^	0.39	9.66 × 10^−14^	Regulation of actin cytoskeleton
nuclear receptor corepressor 2	NCOR2	−1.31	−1.78	−2.94	−1.46	0.41	4.33 × 10^−33^	0.61	4.30 × 10^−18^	Notch signaling pathway
dedicator of cytokinesis 1	DOCK1	−1.28	−2.01	−1.19	−2.12	0.16	1.98 × 10^−41^	0.92	7.24 × 10^−61^	Focal adhesion, Regulation of actin cytoskeleton

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
