# Peer review of "The Protective Effects of Lactoferrin on Aflatoxin M1-Induced Compromised Intestinal Integrity"

_ijms, 2021, doi:10.3390/ijms23010289_

Round 1

Reviewer 1 Report

In this study by Dr. Gao et al. "The protective effect of lactoferrin on impaired intestinal integrity induced by aflatoxin M1" the ability of bovine lactoferrin to protect the intestinal barrier from aflatoxin M1 damage was analyzed both in vitro and in vivo (differentiated Caco-2 cells and Balb/C mice, respectively).

The results obtained showed that treatment with bLF was able to improve the protein expression of claudin-3, occludin and ZO-1, as well as to reduce the increase in intestinal permeability induced by AFM1.

General Comments:

The paper is very interesting but has some flaws.

English in particular needs to be revised.

References are inaccurate.

It would be interesting if the authors described their previous article on the protective effect of lactoferrin on aflatoxin M1-induced toxicity and damage in Caco-2 cells (Zhao et al., 2018) together with the article of Zhao and co-workers (2019) about lactoferrin effect on barrier function of Caco-2 cells. These articles are not mentioned and should be added in references.

Zheng N, Zhang H, Li S, Wang J, Liu J, Ren H, Gao Y. Lactoferrin inhibits aflatoxin B1- and aflatoxin M1-induced cytotoxicity and DNA damage in Caco-2, HEK, Hep-G2, and SK-N-SH cells. Toxicon. 2018 Aug;150:77-85. doi: 10.1016/j.toxicon.2018.04.017. Epub 2018 May 26. PMID: 29753785

Zhao X, Xu XX, Liu Y, Xi EZ, An JJ, Tabys D, Liu N. The In Vitro Protective Role of Bovine Lactoferrin on Intestinal Epithelial Barrier. Molecules. 2019 Jan 2;24(1):148. doi: 10.3390/molecules24010148. PMID: 30609730

Specific comments:

Lines 58-59

“in addition, hLF and bLF exhibit some identical biological functions in various experimental model (s)” (11, 12).

The reference 12 is an article where human lactoferrin is not used.

Lines 59-61

“US Food and Drug Administration (FDA) has approved bLF for practical use due to bLF is generally recognized as safe (GRAS) therefore bLF has been used as a supplement for infant formula (13).

The reference 13 is review about lactoferrin and coronavirus.

Lines 62-63

Considering that bLF is known for its antibacterial anti, nflammatory and immunoregulatory properties (14,15).

The reference 15 is a paper about lactoferrin and arbovirus.

Lines 67-69

The sentence “Only few studies demonstrated that the intestinal injury in epithelial cells and mice could be reversed by LF by through suppressing the inflammation” is not exact. In the literature there are several articles that describe the anti-inflammatory activity of Lf even in patients suffering from Crohn's disease. Some examples:

Tanaka H, Gunasekaran S, Saleh DM, Alexander WT, Alexander DB, Ohara H, Tsuda H. Effects of oral bovine lactoferrin on a mouse model of inflammation associated colon cancer. Biochem Cell Biol. 2021 Feb;99(1):159-165. doi: 10.1139/bcb-2020-0087. Epub 2020 Sep 9. PMID: 32905707

Alexander DB, Iigo M, Abdelgied M, Ozeki K, Tanida S, Joh T, Takahashi S, Tsuda H. Bovine lactoferrin and Crohn's disease: a case study. Biochem Cell Biol. 2017 Feb;95(1):133-141. doi: 10.1139/bcb-2016-0107. Epub 2016 Nov 30.

PMID: 28165294

Hill DR, Newburg DS. Clinical applications of bioactive milk components. Nutr Rev. 2015 Jul;73(7):463-76. doi: 10.1093/nutrit/nuv009. PMID: 26011900

Bertuccini L, Costanzo M, Iosi F, Tinari A, Terruzzi F, Stronati L, Aloi M, Cucchiara S, Superti F. Lactoferrin prevents invasion and inflammatory response following E. coli strain LF82 infection in experimental model of Crohn’s disease. Digestive and Liver Disease. 2014;46(6):496–504. doi: 10.1016/j.dld.2014.02.009.

Line: 278

Please change LY with LF.

Author Response

Dear reviewer,

Thank you very much for providing an opportunity for us to revise our paper, entitled “The protective effects of lactoferrin on aflatoxin M1-induced compromised intestinal integrity” (Manuscript ijms-1483538) for your journal. And thank you so much for the constructive comments and valuable suggestions on the manuscript. According to the valuable advices, we have made a major revision to this manuscript.

We hope that the manuscript could be acceptable for publication in your journal.

Sincerely,

Nan Zheng, PhD

Responds to reviewers' comments:

Reviewer #1:

In this study by Dr. Gao et al. "The protective effect of lactoferrin on impaired intestinal integrity induced by aflatoxin M1" the ability of bovine lactoferrin to protect the intestinal barrier from aflatoxin M1 damage was analyzed both in vitro and in vivo (differentiated Caco-2 cells and Balb/C mice, respectively).

The results obtained showed that treatment with bLF was able to improve the protein expression of claudin-3, occludin and ZO-1, as well as to reduce the increase in intestinal permeability induced by AFM1.

General Comments:

The paper is very interesting but has some flaws.

(1): English in particular needs to be revised.

AU: Thanks for your suggestions. As your suggestion, we have revised the English in the revised manuscript with the help of English editing company. And we have added the related contents in Acknowledgments section of the revised manuscript at Line 474-475 as ‘We also thank International Science Editing (http://www.internationalscienceediting.com) for editing this manuscript’.

(2): References are inaccurate.

AU: Thanks for your suggestions. As your suggestion, we have modified the references in the revised manuscript.

(3): It would be interesting if the authors described their previous article on the protective effect of lactoferrin on aflatoxin M1-induced toxicity and damage in Caco-2 cells (Zhao et al., 2018) together with the article of Zhao and co-workers (2019) about lactoferrin effect on barrier function of Caco-2 cells. These articles are not mentioned and should be added in references.

AU: Thanks for your suggestions. As your suggestion, we added these two related references in Line 70-73 ‘where one study showed that bLF could increase the expression of tight junction proteins and enhance the intestinal epithelial barrier in human intestinal epithelial crypt cells (HIECs) and Caco-2 cells [15].’ and Line 77-78 ‘Although our previous reported study demonstrated that bLF could inhibit AFM1-induced DNA damage in human intestinal Caco-2 cells [20]’ of the revised manuscript.

  1. Zhao, X.; Xu, X.X.; Liu, Y.; Xi, E.Z.; An, J.J.; Tabys, D.; Liu, N. The In Vitro Protective Role of Bovine Lactoferrin on Intestinal Epithelial Barrier. Molecules 2019, 24, 148, doi: 10.3390/molecules24010148.
  2. Zheng, N.; Zhang, H.; Li, S.L.; Wang, J.Q.; Liu, J.; Ren, H.; Gao, Y.N. Lactoferrin inhibits aflatoxin B1- and aflatoxin M1-induced cytotoxicity and DNA damage in Caco-2, HEK, Hep-G2, and SK-N-SH cells. Toxicon 2018, 150, 77-85, doi: 10.1016/j.toxicon.2018.04.017.

Specific comments:

(4): Lines 58-59 “in addition, hLF and bLF exhibit some identical biological functions in various experimental model (s)” (11, 12). The reference 12 is an article where human lactoferrin is not used.

 AU: Thanks for your suggestions. We are sorry for the negligence, and in the revised manuscript, we deleted the reference of 12.

(5): Lines 59-61 “US Food and Drug Administration (FDA) has approved bLF for practical use due to bLF is generally recognized as safe (GRAS) therefore bLF has been used as a supplement for infant formula (13). The reference 13 is review about lactoferrin and coronavirus.

  AU: Thanks for your suggestions. In the reference 13, the content of ‘bLf has been applied in in vitro and in vivo studies, being generally recognized as safe (GRAS) by the Food and Drug Administration (FDA) and available in large quantities’ has been described. As your suggestion, to avoid misunderstanding, we replaced this reference (13) with reference (11, Rosa et al., 2017).

  1. Rosa, L.; Cutone, A.; Lepanto, M.S.; Paesano, R.; Valenti, P. Lactoferrin: A Natural Glycoprotein Involved in Iron and In-flammatory Homeostasis. Int J Mol Sci 2017, 18, doi:10.3390/ijms18091985.

(6): Lines 62-63 Considering that bLF is known for its antibacterial anti-inflammatory and immunoregulatory properties (14,15). The reference 15 is a paper about lactoferrin and arbovirus.

   AU: Thanks for your suggestions. We are sorry for the negligence, and in the revised manuscript, we deleted the reference of 15.

(7): Lines 67-69 The sentence “Only few studies demonstrated that the intestinal injury in epithelial cells and mice could be reversed by LF by through suppressing the inflammation” is not exact. In the literature there are several articles that describe the anti-inflammatory activity of Lf even in patients suffering from Crohn's disease. Some examples:

    AU: Thanks for your suggestions. As your suggestion, we revised the description as ‘There are also several studies demonstrating the anti-inflammatory activity of LF, even in patients suffering from Crohn's disease [16-19]’ in Line 73-76 of the revised manuscript.

  1. Tanaka, H.; Gunasekaran, S.; Saleh, D.M.; Alexander, W.T.; Alexander, D.B.; Ohara, H.; Tsuda, H. Effects of oral bovine lactoferrin on a mouse model of inflammation associated colon cancer. Biochem Cell Biol 2021, 99, 159-165, doi: 10.1139/bcb-2020-0087.
  2. Alexander, D.B.; Iigo, M.; Abdelgied, M.; Ozeki, K.; Tanida, S.; Joh, T.; Takahashi, S.; Tsuda, H. Bovine lactoferrin and Crohn's disease: a case study. Biochem Cell Biol 2017, 95, 133-141, doi: 10.1139/bcb-2016-0107.
  3. Hill, D.R.; Newburg, D.S. Clinical applications of bioactive milk components. Nutr Rev 2015, 73, 463-476, doi: 10.1093/nutrit/nuv009.
  4. Bertuccini, L.; Costanzo, M.; Iosi, F.; Tinari, A.; Terruzzi, F.; Stronati, L.; Aloi, M.; Cucchiara, S.; Superti, F. Lactoferrin prevents invasion and inflammatory response following E. coli strain LF82 infection in experimental model of Crohn’s disease. Digest Liver Dis 2014, 46, 496–504, doi: 10.1016/j.dld.2014.02.009.

(8): Line: 278 Please change LY with LF.

AU: Thanks for your suggestions. We are sorry for this mistake, and we change LY with LF in the Line 326 of the revised manuscript as ‘The underlying mechanisms of the protective effects of LF were clarified by the cross-omics data of the transcriptome and proteome’.

Reviewer 2 Report

In this manuscript the authors report results on the protective effect of Lf on intestinal integrity following aflatoxin M1 treatment in mice and in Caco2 cells, expanding on their previous work on the toxicity of AFM1. While the results may be interesting, a number of issues make the manuscript unsuitable for publication at the present stage. First of all, the manuscript is very confused and English language must be improved: I strongly suggest editing by a native English speaker otherwise the paper is very hard to read. Brief explanations should be added to introduce the results and clarify the experiments presented in most paragraphs (e.g, in par. 2.2 and 2.3 the fact that differentiated Caco2 cells are used should be mentioned in the first sentence).

The Lf and Lf+AFM1 groups are missing and must be added in fig. 5E and 6D.

Experimental details are lacking: transcriptomic and proteomic studies were performed on Caco2 cells at what stage? for how long were treatments with AFM1 and/or Lf performed? 48 h? there are no indications in Results or Methods sections and these details must be provided.

I also see a general over-interpretation of results:

1) in fig 1 Lf restores Cit levels only, IFABP and D-Lactate levels are not much  improved in the Lf+AFM1 group compared to AFM1

2) par 2.2, 2.3 and 2.4, in fig. 4 only very modest effects of Lf are evident in the Lf-AFM1 group compared to AFM1, in par 2.3 and 2.4 the authors state that the effect of Lf is modest and most of the differences are due to AFM1, so the titles of these paragraphs are misleading 

Lastly, abbreviations should be defined in the Results section or when first mentioned, not in Methods.

Author Response

Dear reviewer,

Thank you very much for providing an opportunity for us to revise our paper, entitled “The protective effects of lactoferrin on aflatoxin M1-induced compromised intestinal integrity” (Manuscript ijms-1483538) for your journal. And thank you so much for the constructive comments and valuable suggestions on the manuscript. According to the valuable advices, we have made a major revision to this manuscript.

We hope that the manuscript could be acceptable for publication in your journal.

Sincerely,

Nan Zheng, PhD

Responds to reviewers' comments:

Reviewer #2:

In this manuscript the authors report results on the protective effect of Lf on intestinal integrity following aflatoxin M1 treatment in mice and in Caco2 cells, expanding on their previous work on the toxicity of AFM1. While the results may be interesting, a number of issues make the manuscript unsuitable for publication at the present stage. First of all, the manuscript is very confused and English language must be improved: I strongly suggest editing by a native English speaker otherwise the paper is very hard to read. Brief explanations should be added to introduce the results and clarify the experiments presented in most paragraphs (e.g, in par. 2.2 and 2.3 the fact that differentiated Caco2 cells are used should be mentioned in the first sentence).

AU: Thanks for your suggestion. As your suggestion, we have revised the English in the revised manuscript with the help of English editing company. And we have added the related contents in Acknowledgments section of the revised manuscript at Line 474-475 as ‘We also thank International Science Editing (http://www.internationalscienceediting.com) for editing this manuscript’.

        In addition, as your suggestion, we added the related contents about the model used in the experiment in the revised manuscript at Line 133 (par. 2.2), Line 150-151 (par. 2.3) and Line 196-197 (par. 2.4).

(1): The Lf and Lf+AFM1 groups are missing and must be added in fig. 5E and 6D.

AU: Thanks for your suggestion. As your suggestion, we added the results of LF and LF+AFM1 treatment in Fig. 5E and 6D in the revised manuscript, and the related description also has been added in the Results section at Line 173-178 and Line 217-219 in the revised manuscript.

(2): Experimental details are lacking: transcriptomic and proteomic studies were performed on Caco2 cells at what stage? for how long were treatments with AFM1 and/or Lf performed? 48 h? there are no indications in Results or Methods sections and these details must be provided.

AU: Thanks for your suggestion. As your suggestion, we added the experimental details in the Materials and Methods Line 422-426 ‘In brief, to reduce the variability of samples, triplicate wells of differentiated Caco-2 cells were combined into one sample. For each treatment, a total of three samples were prepared. In brief, the RNA of the differentiated Caco-2 cells without LF and AFM1 treatment (control) and the cells treated with individual 100 μg/mL LF and 8 μg/mL AFM1, and their combination for 48 h were extracted’ and Line 439-443 ‘To reduce the sample variability, three replicates of harvested differentiated Caco-2 cells were pooled into one sample, and a total of three samples were prepared from each treatment. Differentiated Caco-2 cells without treatment were chosen as the control group. Cells treated with 100 μg/mL LF and 8 μg/mL AFM1 individually as well as their combination for 48 h were considered the treated group’ of the revised manuscript.

(3): I also see a general over-interpretation of results: in fig 1 Lf restores Cit levels only, IFABP and D-Lactate levels are not much  improved in the Lf+AFM1 group compared to AFM1

AU: Thanks for your suggestion. As your suggestion, we have modified the description as ‘The addition of LF significantly restored the Cit level and partly decreased the I-FABP and D-lactate contents in the AFM1 treatment group (Fig. 1B-D)’ in Line 99-101 of the revised manuscript.

(4): I also see a general over-interpretation of results:  par 2.2, 2.3 and 2.4, in fig. 4 only very modest effects of Lf are evident in the Lf-AFM1 group compared to AFM1, in par 2.3 and 2.4 the authors state that the effect of Lf is modest and most of the differences are due to AFM1, so the titles of these paragraphs are misleading 

AU: Thanks for your suggestion. As your suggestion, we modified the titles of par 2.2, 2.3 and 2.4 as ‘Effect of LF on injured differentiated Caco-2 cells induced by AFM1’ , ‘Effect of LF on the gene expression in AFM1 treatment in vitro’ and ‘Effect of LF on the protein expression in AFM1 treatment in vitro’ in Line 130, 150 and 195 of the revised manuscript.

(5): Lastly, abbreviations should be defined in the Results section or when first mentioned, not in Methods.

AU: Thanks for your suggestion. We are sorry for this mistake. As your suggestion, we have defined the abbreviations in the Results Line 97, 98, 133, 136, 152, 164, and 198 of the revised manuscript.

Round 2

Reviewer 2 Report

The authors have addressed all my comments.